# Determination and Comparison of Physical Meat Quality Parameters of Percidae and Salmonidae in Aquaculture

**DOI:** 10.3390/foods9040388

**Published:** 2020-03-27

**Authors:** Katrin Komolka, Ralf Bochert, George P. Franz, Yagmur Kaya, Ralf Pfuhl, Bianka Grunow

**Affiliations:** 1Institute of Muscle Biology and Growth, Leibniz Institute for Farm Animal Biology (FBN), 18196 Dummerstorf, Germany; 2Mecklenburg-Vorpommern Research Centre for Agriculture and Fisheries (LFA MV), Institute of Fisheries, Research Station Aquaculture, 18375 Born, Germany

**Keywords:** meat quality, shear force, rainbow trout, maraena whitefish, pikeperch, European perch

## Abstract

Although aquaculture has been the fastest growing food sector for decades, there are no standardized parameters for most of the fish species regarding physical meat quality. Therefore, this study provides for the first time an overview of the physical meat characteristics of the most important fish species of the German Baltic Sea coast. Traditional farmed salmonids (rainbow trout (*Oncorhynchus mykiss*) and maraena whitefish (*Coregonus maraena*) as well as two percids (European perch, *Perca fluviatilis* and pikeperch, *Sander lucioperca*) were utilized for this comparison. The results demonstrate that the meat of the salmonids is very analogous. However, the *post mortem* degradation process starts faster in trout meat. In contrast, the meat quality characteristics of the percids are relatively different. The meat of pikeperch has comparatively low shear strength with a high water-holding capacity resulting in high meat tenderness. The opposite situation is present in European perch. The results indicate that it is not possible to establish the overall quality characteristics for fish or production form, as there is a high range of variability. Consequently, it is particularly important that meat quality characteristics are developed for important aquaculture species for further improvement through changes in husbandry conditions when necessary.

## 1. Introduction

Fish is an important source of animal protein for human nutrition. As good meat quality is associated with a healthy animal, quality analysis can improve the marketability of aquaculture products. Therefore, the comprehension of how muscular properties affect fish meat quality can be of great importance in the aquaculture industry.

So far, analyses of fish meat are mostly limited to nutrients, proteins and lipids, their contents, and their compositions [1,2]. There are also efforts to increase product quality and productivity through genetic analyses. However, the basic methods used to check meat quality in mammals and birds are still not standard in fish due to its high species diversity.

Meat quality can be defined by the use of various characteristics. In the industry, these include not only the nutritional–physiological value, but also the processing as well as sensory and hygienic–toxicological parameters [3]. In contrast, the consumer′s quality assessment is based on subjective criteria such as freshness, appearance, odor, aroma, and taste. Further objectively measurable factors that are important for the consumers are texture, firmness, juiciness and moisture, water-holding capacity, drip loss or cook loss, pH value, and color [4]. The texture is one of the main quality parameters for fresh fish and is defined by its dryness, chewiness, and juiciness [5]. In the fish industry, the texture is commonly tested by the “finger method”, which to a large extent depends on the subjective evaluation of the person who does the test [6]. For that reason, Sigurgisladottir and colleagues studied already different methods for examining the texture and could show that the shear force method, whose use is standardized for analyzing the meat of mammals and birds, was found to be more sensitive than the puncture methods and best suited for practical application [7]. Nevertheless, this method is still not standardized for fish. One of the causes is the high diversity of the fish and the associated texture values. Apart from the inconsistent measurement methods, there is no study available that gives an overview of different fish species in their meat quality with regard to the physical parameters. Most of the studies usually focus only on one species to determine possible differences in fish meat in relation to fish keeping temperatures, food, slaughtering method, and product storage [8,9,10,11,12,13,14] or cook loss regarding water-holding capacity in fish species, and methods of cooking [15,16,17].

The next problem is the lack of information regarding processing parameters. The processing parameters of meat quality are hereditary and can therefore be influenced to a considerable extent by breeding and treatment of the animals in the pre- and perimortem period and the storage of the carcasses in the *post mortem* period [18,19,20,21]. There are several studies for mainly Atlantic salmon or other salmonid species that investigate the effects of stress by handling or slaughter methods on the rigor mortis process and in some studies also regarding muscle parameters [22,23,24,25,26,27,28,29,30]. Analysis in Atlantic salmon exhibited that the meat of stressed fish became stiffer, and the onset of rigor occurred about one day earlier than in rested fish [24]. In contrast to this study, Sigholt and colleagues describe that the meat of a stressed fish had a lower score for texture compared to stress-free fishes after a storage of 9 days at cold temperatures (0.4–3 °C) [28]. In total, it is known that pre-slaughter and slaughter stressful practices could have an important effect on the fish flesh quality and in this context especially the physical parameters [31]. A stressed fish produces more muscular energies and therefore exhibits a higher lactic acid content, which reduces the muscular pH, which in turn increases the rate of rigor mortis onset. In this way, slaughter methods could have significant negative effects on the technological traits, flesh quality, and keeping quality of fish. It seems that asphyxia and electrically stunned fish are more stressed than spiked, knocked, and live chilled fish. A carbon monoxide application leads to a stress-free fish due to the anaesthesia, but it could be proven that these methods also influence the meat quality. Due to anaesthesia, muscle relaxation is increased, and the physical parameters do not reflect the actual values. Additionally, the fillet color significantly increases in redness and lightness [32]. To maintain the best original quality, the fish should be stunned until death and killed without any avoidable stress [31]. Besides some general studies of the killing method in rainbow trout, there is still a huge lack of information about the fish meat quality for the economically important fish species of Germany. Moreover, in these studies, information is missing. For farmed fish, only in case of Atlantic salmon, a 283-day (d) grow-out period in recirculating aquaculture systems (RAS) is known [33], whereas farmed fish from unknown production systems were used in other studies [34,35,36]. All these information underline the importance of analyzing the physical fish flesh quality of a stress-free killed fish of diverse species that were grown in a controlled aquaculture production. Consequently, the aim of this study was to provide an overview of the physical quality parameters that influence the consumer′s purchase decision.

We would provide first data on the meat quality of new aquaculture candidates from RAS. For the establishment of suitable methods to analyze fish meat, the known physical methods of meat analysis used for other farm animals, such as cattle and swine, were applied and transferred. To validate these methods, four species were used, rainbow trout (*O. mykiss)* from a traditional flow-through system, and maraena whitefish (*C. maraena*), pikeperch (*S. lucioperca)*, and European perch (*P. fluviatilis*) from RAS. Main focuses in the consideration of consumer relevance were set on marketable sized fishes and on species that are new for aquaculture production [36,37,38,39,40]. The age of the fishes was not considered, as it is irrelevant for sale. It is known that fish at a fully developed stage convert food not only into protein deposition but also into fat [41]. For economic reasons, fishes in aquaculture and therefore also our used fishes were used when reaching the commercial weight. As European perch and pikeperch are traded in a higher price class and therefore also weight class, as breeding and rearing is more difficult, the percids were older than the salmonids.

However, the aim of this study was precisely to investigate the physical quality of meat in the most important and commercially available fish species. With this study, we want to achieve that existing guidelines for meat and meat products equally be valid for fish species in order to guarantee quality for the consumer.

## 2. Material and Methods

### 2.1. Fish Rearing and Experimental Design

All fishes were provided from a research station for aquaculture, the Institute of Fisheries of the State Research Centre for Agriculture and Fishery Mecklenburg-Vorpommern in Born (Germany). In this study, 15 adult animals of both sexes from each *P. fluviatilis* (PFL), *S. lucioperca* (SLU), *O. mykiss*, strain BORN (OMY) [42], and *C. maraena* (CMA) were used (Figure 1). Since aquaculture products are traded according to size and weight, these criteria were also decisive for the selection of animals in our study. Therefore, the animals were kept under the most commercial and economically efficient aquaculture conditions, which are standard conditions in the region of northeast Germany.

OMY: Rainbow trout (BORN strain) hatched in March 2018 were kept in an indoor flow-through system with natural water from the adjacent Darss-Zingst Bodden chain, salinity 3 to 5 PSU, seasonal water temperature changes (0.5 to 26 °C), and natural light conditions except from September to March, where artificial lighting was maintained 8 h light–16 h dark. Fish were regularly sorted and sized to adequate rearing units and stocking densities during their life span. OMY were finally sorted on 8 April 2019 and stocked at a density of about 16 kg m^−3^ in 5 m³ round tanks weighting 303 g in mean. Fishes were maintained (water temperature 9–15 °C) on a commercial 4.5 mm salmonid pellet diet applied by automated feeders (Aller Aqua Silver, crude protein (CP) 43%, crude fat (CF) 22%, nitrogen-free extract (NFE) 20%, ash 7%, fiber 2%, digestible energy (DE) 19.5 MJ) and a feeding rate of 1%–1.3% body weight until sampling.

CMA: Whitefish hatched in February 2017 as F1 aquaculture generation were kept according to the size in different rearing tanks in freshwater RAS. After last sorting on 28 April 2018, CMA were reared in RAS of a total volume of 50 m³ containing 10 round tanks each 3 m³ in size, which were stocked at a final density of about 8 kg m^−3^. Fishes were fed a commercial diet (Aller Aqua Thallassa Ex 2 mm pellet size, CP 48%, CF 15%, NFE 19%, ash 7.4%, fiber 2.6%, digestible energy (DE) 18.2 MJ) at a feeding rate of 0.7% by automated feeders. Water temperature was maintained at 20.9 ± 1.8 °C and 24 h lighting was applied during the period. Water quality was assured by continuous purification and disinfection (drum filter, moving bed bio-filter and UV light). Water parameters including NH^4+^, NO^2-^, and NO^3-^ concentrations were measured weekly, while pH value, temperature, and oxygen saturation were constantly recorded. Dissolved oxygen was maintained >8 mg L^−1^ and all physical water parameters were kept in an optimal range.

PFL: Eggs from wild animals were collected in spring 2016 and hatched in the research station. PFL were kept according to the size in different rearing tanks in freshwater RAS. After last sorting on 22 April 2018, animals were reared in similar RAS as described above and stocked at a final density of about 45 kg m^−3^. Fishes were fed a commercial diet (Coppens Repro 8 mm pellet size, CP 48%, CF 13%, ash 6.3%, fiber 1.2%, digestible energy (DE) 16.7 MJ) at a feeding rate of 0.8% by automated feeders. Water temperature was maintained at 22.8 ± 1.7 °C, and 24 h lighting was applied during the period. All the water quality, see above, was kept in an optimal range.

SLU: Pikeperch hatched in May 2017 after an artificial reproduction of parents from aquaculture broodstock were kept according to the size in different rearing tanks in freshwater RAS. After last sorting on 21 August 2018, animals were reared in the same RAS as that of PFL described above and stocked at a final density of about 16 kg m^−3^. Fishes were fed a commercial diet (Coppens Supreme-10 4.5 mm pellet size, CP 49%, CF 10%, ash 9.4%, fiber 1.8%, digestible energy (DE) 15.5 MJ) at a feeding rate of 1.0% by automated feeders. All the water quality, see above, was kept in an optimal range.

Animals were transported alive in a transport box with an additional oxygen provision from the Institute of Fisheries in Born to the Leibniz Institute for Farm Animal Biology (FBN) in Dummerstorf, Germany. Transport was performed in groups of three to five animals due to animal welfare. The fishes were not fed for two days before transport. The sampling and processing of each species was carried out within three days to two weeks. All 60 specimens were processed within two months (April–May 2019).

### 2.2. Sampling and Morphometric Data Collection

According to the German Animal Welfare Act (§ 4(3) TierSchG), fishes were stunned by a beat on the head and were directly killed by bleed cut in the heart as well as cutting of the spinal cord posterior to the head. Morphometric parameters (total length, circumference, and total weight) were measured. To reduce individual differences in the filleting procedure, a single person prepared all the fillet samples. Photos of each fish were taken with a Canon EOS 4000D. Images were optimized with Adobe Photoshop CS 4 (Adobe Inc., San Jose, CA, USA) by adjusting the tonal range as well as the brightness and global contrast.

### 2.3. Physical Meat Quality

The meat quality analysis was performed immediately after death (first measurements needs to be performed 5 min after death) on the right fish fillet without skin, scales, and bones from 15 animals of each species. The fillet quality was determined by measuring the shear force, water-holding capacity, pH, electrical conductivity, and impulseimpedance in the muscle tissue.

The tenderness of the fillet was determined by measuring the shear force 30 min *post mortem*. The shear force was recorded by a Texture Analyser TA.XTplus (Winopal, Elze, Germany) with a Warner–Bratzler blade (USDA Standard, 1.2 mm thickness, 60° angle with 4.5 mm/sec). The Warner–Bratzler method is the state of the art in measuring the shear force of food materials and standard method in meat science in mammals and other species. This has been well documented by many publications in the past [43]. This shear force measurement was also determined to be the best method for analyzing fish meat texture [7]. In comparison to the meat of mammals and poultry, fish meat has a different and a softer structure. We considered this by adapting the measuring procedure. While the protocols for red and white meat in mammals instruct cooking before probing, we examined the raw fresh fish fillet because the cutting resistance of raw fish is bigger than that of cooked fish fillet. So, any individual differences in the shear force will be rather seen in raw rather than in cooked fillet. For better repeatability, three cuts of the right fillet with 4 × 8 cm were performed across to the muscle fiber orientation and parallel to the septum at room temperature.

The water-holding capacity (WHC) was examined with the filter press method via Hypress [44]. This method was established in 1975 in Dummerstorf and is a further development of the traditional method of GRAU & HAMM [45], which is also a standard method in meat science. Three fresh fillets of 0.3 ± 0.05 g were cut out and each was placed on a 6 × 6 cm filter paper that was prepared with a 4 cm circle of varnish. The varnish preparation will ensure the complete removal of the pressed meat before weighing. The samples on the filter paper were pressed 5 min in the Hypress device, which ensures a constant pressure of 35 kg during the measurement process. After the pressing, the meat was peeled off the filter paper, which was then weighed again. The weight difference was recorded and calculated using the following formula: $$\mathrm{WHC}\;\left[\%\;\right]=\frac{\left(\mathrm{initial}\;\mathrm{sample}\;\mathrm{weight}\;\left[g\right]-\mathrm{after}\;\mathrm{pressure}\;\mathrm{sample}\;\mathrm{weight}\;\left[g\right]\right)\times 100\;\%}{\mathrm{initial}\;\mathrm{sample}\;\mathrm{weight}\;\left[g\right]}$$ WHC is expressed in percent of the initial weight of the sample.

The three parameters—pH, impulse impedance, and electrical conductivity—were measured once for each of every animal fillet 5 min and 1 h *post mortem.* At both time points, the temperature of the fillet was 18 °C, which was measured by a puncture thermometer Testo 926 (Testo, Titisee-Neustadt, Germany) diagonally at the thickest point of the fillet to ensure a correct core measurement.

The pH value is by far the most important meat quality parameter in all species. It gives information about the chemical development of meat and the ripening process in mammals and poultry. This is also the case for fish.

The detection of pH values was performed by pH-Star (Matthäus, Eckelsheim, Germany). This device is one of the finest puncture pH electrodes for field testing on the market.

The electrical conductivity is also an important meat quality parameter in farm animals. It gives information about the cell degradation *post mortem* and therefore about the ripening process and the development of flavor components. The impulse impedance is very similar to the electrical conductivity, but instead of a fixed frequency, the impulse impedance measures the dielectric loss factor over an array of frequencies, which is an indicator of intact muscle cells [46]. These methods can also contribute to the detection of meat and fish spoilage [47].

Electrical conductivity was analyzed using an LF-Star (Matthäus, Eckelsheim, Germany). Impulse impedance was performed by a Meat Check 150 (sigma electronic, Erfurt, Germany), respectively. Both systems have double electrodes, which need sufficient physicochemical “surrounding“ to work properly. This was ensured as described above.

The color was recorded with a CR-300 Chroma Meter (Minolta, Weilheim, Germany) using standard Illuminant D65 and a 2° standard observer according to the L*a*b* CIELAB system. The L*-value describes the lightness between 0 (black) and 100 (white). The a*b* coordinate plane was constructed by applying the counter color theory. On the a* axis, green and red are opposite to each other, while the b* axis runs between blue and yellow. Complementary color tones face each other by 180°, gray is in their center (the coordinate origin a = 0, b = 0). The L*a*b* measurement is also an approved method in meat science and is well documented. Coloration of the fish fillet was measured as the surface color on the inner face of the fillet 10 min *post mortem.* Lightness, redness, and yellowness values were gained as the mean from three measure points each of the dorsal and ventral muscles along the horizontal septum.

### 2.4. Statistics

Statistical analysis was carried out using SAS software version 9.2 (Statistical Analysis institute Inc., USA). Data of the morphometric and quality analysis were displayed in mean ± S.E.M. (standard error of the mean) of 15 animals per species and analyzed via one-way ANOVA (significance level *p* < 0.05). The fish species comparisons were done by using Tukey′s pair-wise comparison test. For comparison of the two time points of pH, EC, and impulse impedance, a paired t-test procedure was used. Results are displayed in means ± S.E.M. for each fish species with a significance level of *p* < 0.05. Means with different superscript letters differ significantly between the animal groups (*p* < 0.05). Significant differences within one fish species over time are shown as * *p* < 0.05, ** *p* < 0.01, and *** *p* < 0.001. Graphs were created by Graphpad Prism 8 (Graphpad Software, San Diego, CA, USA).

## 3. Results

### 3.1. Animals

As demonstrated in Figure 1, the fish used in this study displayed a huge variety in their morphometric parameters. The marketable size of salmonids maraena whitefish and rainbow trout were with a length of 32.95 ± 0.27 cm and 31.11 ± 0.24 cm smaller than the two representatives of the examined Percidae. The European perch had an average length of 37.58 ± 0.37 cm and the pikeperch had an average length of 49.53 ± 0.85 cm. The total weight of percids 777.07 ± 22.00 g (PFL) and 994.33 ± 63.12 g (SLU) has been higher than in both salmonid species (Table 1).

### 3.2. Flesh Quality

Concerning the shear force and the water-holding capacity, there were no significant differences in the salmonid species reared in different culture systems (Figure 2A,B). The shear force measurement of the marena whitefish was 24.70 ± 1.26 N and that of the rainbow trout was 28.98 ± 1.16 N (Figure 2A). In European perch, the shear force was significantly higher (53.89 ± 1.76 N), and in pikeperch, it was significantly lower (15.60 ± 0.68 N) compared to the salmonids (Figure 2A). In the analyses, the water-holding capacity of pikeperch was particularly noticeable. On the one hand, the values had a larger dispersion, and on the other hand, these values were significantly higher than for the other three species (CMA: 20.90 ± 1.26%, OMY: 27.53 ± 0.75%, PFL: 16.45 ± 0.60%, and SLU: 55.43 ± 6.05%, Figure 2B).

The initial pH values ranged between pH 6.6 and 7.0 (Figure 2C) in all specimens. The comparisons between the 5 min and one hour measurements showed a highly significant decrease. After one hour, the pH values were 4% to 6% lower (Figure 2C).

When measuring the impulse impedance, significant changes over time could only be detected for trout (5 min: 63.40 ± 1.25%, 1 h: 55.20 ± 1.57%) and pikeperch (5 min: 72.87 ± 1.60%, 1 h: 65.07 ±1.96%). A comparison between all species showed that the impulse impedance of the muscle meat in the salmonids is significantly lower compared to that in the percids (Figure 2D).

Electrical conductivity was similar for salmonids and pikeperch within the first five minutes after death (OMY: 4.45 ± 0.12 ms, CMA: 4.05 ± 0.17 ms, SLU: 4.03 ± 0.19 ms), whereas it was significantly higher for perch muscle meat by 8% to 18% (Figure 2E). One hour *post mortem*, no significant differences were detectable between the species. However, in the rainbow trout, significantly increased conductivity values were observed one hour *post mortem* (5 min: 4.45 ± 0.12 ms, 1 h: 5.19 ± 0.21 ms), Figure 2E.

Instrumental analysis of color parameters showed differences between the salmonids and the percids. Maraena whitefish had a lightness of 43.64 ± 0.83, a redness of 5.21 ± 0.29, and a yellowness of 4.59 ± 0.47, which was similar to rainbow trout with values of 42.27 ± 0.74 (lightness), 4.51 ± 0.31 (redness), and 4.53 ± 0.32 (yellowness), as shown in Table 2. Significant differences were especially found in pikeperch, in which higher L* values of 48.23 ± 0.51 indicate a lighter coloration of the meat. Additionally, less red color was present in the pikeperch fillets (0.89 ± 0.13, Table 2).

## 4. Discussion

In the investigation of the quality of fish meat, the physical parameters have been found to have less importance in aquaculture research and product quality so far due to large species differences. For this reason, the fillets of four marketable sized species from two different families, reared either in a flow-through system or RAS, were examined in this work with regard to the parameters that are firmly established for mammals. In order to minimize the impact of husbandry conditions on meat quality, the RAS animals came from the same aquaculture facility. Additionally, the slaughter process was the same for all specimens.

The texture, which is an important property of fish muscle, is determined by the shear force [48]. In order to objectify the firmness of the meat, measurements of the shear forces according to Warner–Bratzler [49] were carried out to indicate the maximum force required to divide meat according to defined conditions. There is a negative correlation between the shear values and the evaluation of tenderness [49]. Basically, two protein structures in the muscle have a decisive influence on the strength of the meat: the connective tissue and the structure of the myofibrils [50,51]. For mammals, the degradation of the proteins is well studied. In fish, it is known that the *post mortem* degradation and its mechanisms differ among the species. For example, desmin is degraded *post mortem* in sardine and turbot, but not in sea bass and brown trout [52]. Moreover, the proteolytic processes that finally lead to the meat softening could be associated with the activity of some other endogenous proteases, such as trypsin-like enzyme, collagenase-like enzyme, and metalloproteases [53]. Furthermore, tenderness is clearly connected with water losses. Our analyses showed that both salmonid species have similar values for both shear force and water-holding capacity. Analyses of the percid species, on the other hand, showed significant differences, both compared to salmonids and within the family. European perch has the highest shear force value with 53.9 ± 1.8 N and the lowest water-holding capacity with 16.4 ± 0.6 % (Figure 2A,B). Therefore, the meat of the European perch is not as tender as that of the other species. The fillet of pikeperch had the lowest shear forces, which were even more than 10 N lower than those of the salmonids. Therefore, the tenderness of this meat is very high. This low value is caused among other things by the high water-holding capacity in the pikeperch meat, which had the highest values and also a very high variability compared to the other three fish species. It needs to be investigated whether another food source could possibly reduce the water content in the meat and the high variability, especially as the pikeperch meat is traded in a higher price class. The fact that the meat quality can be changed by the feed has already been proven in other species. For example, Red tilapia had a higher weight if the feed exhibit higher protein contents [54]. The influences of different protein diets were also analyzed in rainbow trout. Fillets from rainbow trout were evaluated as softer when fed with a high-lipid diet compared to fillets from fish fed a low-lipid diet. Nevertheless, there is still a huge lack of information about the role of fish nutrition on the texture of fish fillets [6].

The next step was to examine the pH value and the electrical properties over time, as both parameters have a significant impact on the resulting tenderness of meat [55]. After death, the blood circulation is interrupted, and hence, the provision of oxygen to the muscle tissue is stopped. Consequently, the degradation of glycogen to lactic acid occurs exclusively in an anaerobic manner [56]. When lactic acid accumulates, the pH value of the muscle drops down. Our analysis showed that in all four examined fish species, the mean pH value has been between 6.7 and 7.0 five minutes after slaughtering, which is typical for living fish muscles [57]. An hour later, the values decreased significantly by around 4% to 6%, showing the higher amount of lactate due to glycolysis. In general, fish meat has less glycogen than in mammalian meat, and therefore, its pH decreases in a lower rate *post mortem* [58]. In gilthead sea beam, Kyrana and colleagues examined that even four hours after slaughter, the pH was around 6.2 [59]. In other fish species such as halibut or tuna, pH values of 5.4 to 5.6 could be found [60]. However, most of the fishes show an ultimate pH value of around 6.0 to 6.2 [58]. The time for the beginning of the *rigor mortis* differs also among fish species. For example, at 10 °C, in Japanese flounder (*Paralichthys olivaceus*), it takes 6 h and in carp (*Cyprinus carpio*), it takes 60 h [61]. The long-term examination of the pH value, including the *rigor mortis*, will be the goal of a future study. As a part of this, the start time point of the pH value increase will be determined, reflecting the production of alkaline bacterial metabolites in spoiling fish.

According to Poli and colleagues, the stress level can also be derived from the pH value. Acute stress is associated with increased muscle activity and thus with anaerobic glycolysis, which in turn leads to a lower pH level [31]. Therefore, the development of rigor is much faster in stressed than in unstressed fish [24]. During longer lasting stress, the lactic acid production is stopped, and the energy reserves are used. Therefore, animals under persistent stress show no changes in pH over a prolonged period after slaughter [58]. In this study, all four species exhibited a highly significant decrease of the pH value one hour after slaughter. This allows the assumption that the stress level in the examined fishes was low.

In addition to the pH, the electrical properties of the meat also indicate a change after death. One cause is the change in the intra- and extracellular electrolyte balance. The impulse impedance decreases, and in contrast, the electrical conductivity increases, because the cell membranes get destroyed and more ions are released [62]. In addition, species variations are present in this regard. The European perch with the highest shear force values showed the smallest differences over time. The electrical conductivity only changes by 2%, and the impulse impedance changes by 3.7%. This indicates that the *post mortem* process is slowest in the European perch. In contrast, it can be said that in the rainbow trout, the *post mortem* process was most rapid, as both impulse impedance and WHC showed significant differences within the one-hour period of the analysis. In particular, the changes in the impulse impedance are considerable at 12.5% to 15.5%.

Besides these factors, the color also plays an important role for the consumer, because unexpected colors are unattractive and even repulsive [63]. To improve the color in some salmonid species such as the *Salmo salar*, carotenoid astaxanthine are supplemented to the feed [41]. In our used species, these additives are not used; deductively, we examined the natural flesh color. In addition, it is unknown to what extent this type of color additives influence the natural color change of the fillet over time. For that reason, we examined the properties of the white muscle meat in all four species. In pikeperch, the meat was lighter, which could be due to the high water amount and the lower texture of the meat. In future studies, for these species, the influence of storage procedure and time on the meat color has to be determined. The impact of storage and also stress on tissue coloration was already shown for other species [64,65]. In addition, Robb and colleagues showed the influence of chemical anaesthesia and electrostimulation on the meat color of rainbow trout [66]. The meat lightness of electrostimulated fish is lower and continues to decrease during the first 30 h, whereas the meat lightness from anaesthetized animals continues to increase during the first hours. As far as we know, there are no data about the meat color of rainbow trout without having the influence of electric stimulations or chemicals and also not for the other three species used in this study.

## 5. Conclusions

Objective food quality only differed slightly between rainbow trout, from a traditional flow-through system with high water exchange, and maraena whitefish from RAS, with restricted water exchange rates and purification technology used in recirculating water, as the measured parameters were mostly similar to each other. However, regarding decomposition, it was found out that the meat from rainbow trout was more rapidly affected by *post mortem* degradation. Although both pikeperch and European perch belong to the same family, and were kept in the same RAS for several months, their meat quality was surprisingly different. Our research and also our study show that the physical meat quality of fish should no longer be neglected and that standards should also be introduced here.

## Figures and Tables

**Figure 1 foods-09-00388-f001:**
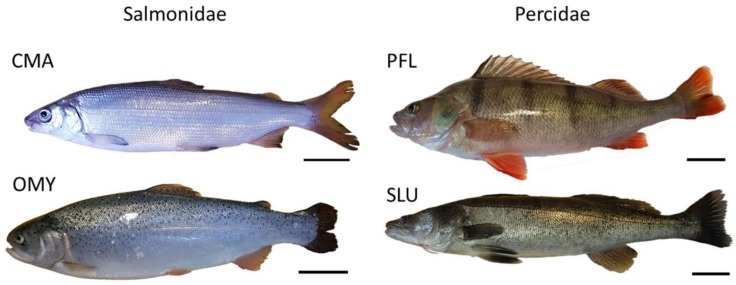
Illustration of the species used in this study. 15 fishes of two salmonid species *Coregonus maraena* (CMA) and *Oncorhynchus mykiss* (OMY) as well as two perciforms *Perca fluviatilis* (PFL) and *Sander lucioperca* (SLU) were analyzed. Scale: 5 cm.

**Figure 2 foods-09-00388-f002:**
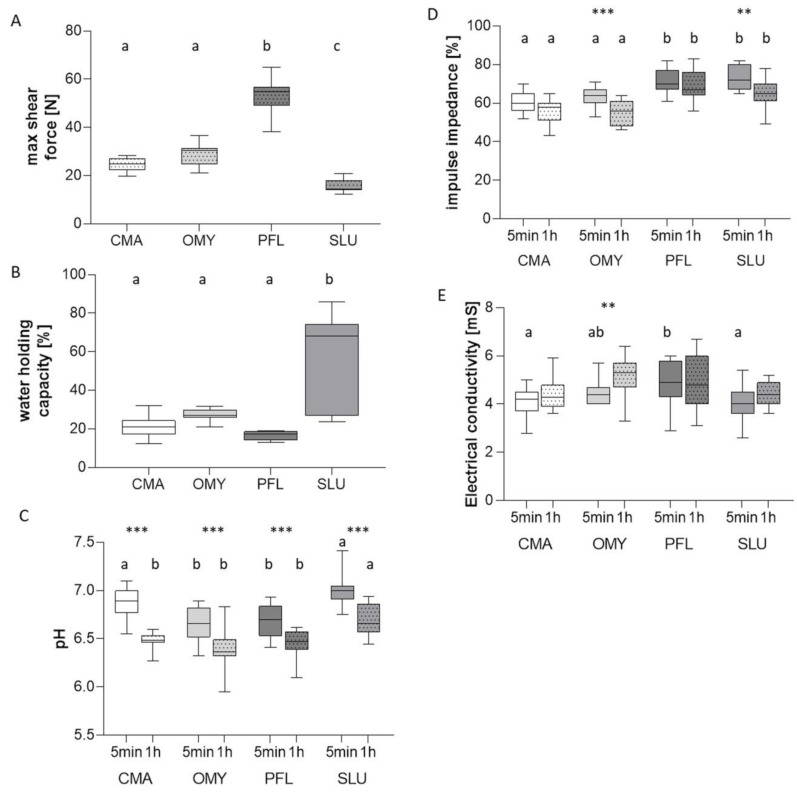
Overview of fish flesh quality analysis. Box–whisker plot of maximal shear force (**A**), water-holding capacity (**B**), pH value (**C**), impulse impedance (**D**) and electrical conductivity (**E**) are illustrated for *Coregonus maraena* (CMA) and *Oncorhynchus mykiss* (OMY) as well *Perca fluviatilis* (PFL) and *Sander lucioperca* (SLU). PH value (**C**), impulse impedance (**D**), and electrical conductivity (**E**) were measured five minutes and one hour after slaughter. Means with different superscript letters differ significantly between the animal groups (*p* < 0.05). Additionally significant differences within one fish species over time are shown as * *p* < 0.05, ** *p* < 0.01 and *** *p* < 0.001.

**Table 1 foods-09-00388-t001:** Overview of morphometric parameters. The mean ± S.E.M. of the total length, circumference, and total weight of 15 fishes from two salmonid species *Coregonus maraena* (CMA) and *Oncorhynchus mykiss* (OMY) as well as two perciforms *Perca fluviatilis* (PFL) and *Sander lucioperca* (SLU) were measured.

	CMA	OMY	PFL	SLU
	Mean ± SEM	Mean ± SEM	Mean ± SEM	Mean ± SEM
total length [cm]	32.95 ± 0.27	31.11 ± 0.24	37.58 ± 0.37	49.53 ± 0.85
circumference [cm]	16.00 ± 0.21	19.31 ± 0.26	25.17 ± 0.30	22.96 ± 0.56
total weight [g]	327.07 ± 7.59	434.93 ± 10.79	777.07 ± 22.00	994.33 ± 63.12

**Table 2 foods-09-00388-t002:** Overview of color parameters. Mean and S.E.M. of L*: lightness, a*: redness, and b*: yellowness for the fillets of *Coregonus maraena* (CMA) and *Oncorhynchus mykiss* (OMY) as well *Perca fluviatilis* (PFL) and *Sander lucioperca* (SLU). Means with different superscript letters differ significantly between the animal groups (*p* < 0.05).

	Salmonidae		Percidae	
	CMA	OMY	PFL	SLU
	Mean ± SEM	Mean ± SEM	Mean ± SEM	Mean ± SEM
L*	43.64 ± 0.83 ^a^	42.27 ± 0.74 ^a^	42.71 ± 0.58 ^a^	48.23 ± 0.51 ^b^
a*	5.21 ± 0.29 ^a^	4.51 ± 0.31 ^a^	3.37 ± 0.27 ^b^	0.89 ± 0.13 ^c^
b*	4.59 ± 0.47	4.53 ± 0.32	5.44 ± 0.32	4.58 ± 0.28

L*: lightness (L* = 0 black and L* = 100 white), a*: red-green component and b*: yellow-blue component.

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
