# Peer review of "Determination and Comparison of Physical Meat Quality Parameters of Percidae and Salmonidae in Aquaculture"

_foods, 2020, doi:10.3390/foods9040388_

Round 1
Reviewer 1 Report
the paper describes a study of physical meat quality parameters in fish. The authors claim in the abstract that this proves for the first time an overview of... The authors should do a much better literature survey - it may be the first time that these particular fish species has been compared, but there ar a large number of studies on quality of fish - including texture and water holding...
water holding capacity - methods for determinijg this has been developed for fish and was published in the 1980s - but there has also been later development of the method - presented in the literature and also forming the basis for several PhDs.
whay do the authors claim that their methoid for texture determination is better than other methods? Measuring texture after 30 minutes?? then the fish could be entering rigor
Yes I agree that slaughter methods may have a significant influence on fish quality but this has also been studied and the results published
Author Response
the paper describes a study of physical meat quality parameters in fish. The authors claim in the abstract that this proves for the first time an overview of... The authors should do a much better literature survey - it may be the first time that these particular fish species has been compared, but there ar a large number of studies on quality of fish - including texture and water holding...
Thank you for pointing out that our statements have been imprecise here. We have now adjusted them and included instead of “…fishes…” to “….most of the fish species…”
water holding capacity - methods for determinijg this has been developed for fish and was published in the 1980s - but there has also been later development of the method - presented in the literature and also forming the basis for several PhDs.
We had added this physical factor in more detail in the introduction. We are aware of the fact, that publications are available about water holding capacity. However, in most of them a cooking method was analysed and that’s why there is no bigger comparable study of different fish species reared under the same conditions.
whay do the authors claim that their methoid for texture determination is better than other methods? Measuring texture after 30 minutes?? then the fish could be entering rigor
Thank you for this comment. It was not our intention to claim. We only wanted to mention that already in 1999 Sigurgisladottir et al. could prove that the shear force measurement (like we used in this present study), used for mammals and birds can be also used for fish. This method is more objective compared to the still used “finger method” and provide clear results. – please see line 64-70
The time delay of the tenderness measurement resulted from the working routine, whereby priority was given to the more sensitive measurement of pH, electrical conductivity, impulse impedance and the color. The certainty of no rigor mortis occurred in the fillet during the duration of the measurements was reliably achieved by repeated data acquisition 1 h post mortem.
Yes I agree that slaughter methods may have a significant influence on fish quality but this has also been studied and the results published
Thank you for your comment. We agree and for that reason, this was not part of our study. We just did mention that the slaughter method has an impact on fish quality and various examples are given in the introduction. To eliminate the factor of slaughter method, we have used the same procedure in all four species.
Reviewer 2 Report
I recommend changing the citation style for the references below, as with other sources.Line 66: For that reason, Sigurgisladottir et al., (1999) studied already different methods for examining the texture.....
Line 78: Already 2005, Poli et al. described that a pre-slaughter and slaughter stressful...
Line 92: ...only in case of Atlantic salmon a 283 d grow out period in RAS is known [21]... I recommend explaining the abbreviation RAS = Recirculating aquaculture systems (RAS); 283 d = 283 day
Lines 233-234: Brightness and Color values were gained as mean ± S.E.M. from three measure points of the on the dorsal and...
L* = Lightness, a* and b* = colour coordinates or rednes and yellowness.
Line 390: ...In addition, Robb et al. (2000) showed the influence of.... I recommend changing the citation style for the reference with number style.
Flesh Quality should be described better for clarity through the values in the tables. The Figure 1 is not ideal for comparing values with other authors and publications.
Author Response
I recommend changing the citation style for the references below, as with other sources.
Line 66: For that reason, Sigurgisladottir et al., (1999) studied already different methods for examining the texture.....
Thank you for this advice – we did change to “….Sigurgisladottir and colleagues….”
Line 78: Already 2005, Poli et al. described that a pre-slaughter and slaughter stressful...
We did change to “….Poli and colleagues….”
Line 92: ...only in case of Atlantic salmon a 283 d grow out period in RAS is known [21]... I recommend explaining the abbreviation RAS = Recirculating aquaculture systems (RAS); 283 d = 283 day
Thank you for the hint to explain all abbreviation. We have included the mentioned abbrevations.
Lines 233-234: Brightness and Color values were gained as mean ± S.E.M. from three measure points of the on the dorsal and...
L* = Lightness, a* and b* = colour coordinates or rednes and yellowness.
We did change this section and explained it in more detail.
“The color was recorded with a CR-300 Chroma Meter (Minolta, Germany) using standard Illuminant D65 and 2° standard observer according to the L*a*b* CIELAB system. The L*-value describes the lightness between 0 (black) and 100 (white). The a*b* coordinate plane was constructed by applying the counter color theory. On the a* axis green and red are opposite each other, the b* axis runs between blue and yellow. Complementary color tones face each other by 180°, grey is in their center (the coordinate origin a*=0, b*=0).
Line 390: ...In addition, Robb et al. (2000) showed the influence of.... I recommend changing the citation style for the reference with number style.
We did change to “….Robb and colleagues….”
Flesh Quality should be described better for clarity through the values in the tables. The Figure 1 is not ideal for comparing values with other authors and publications.
Thank you for the advice. The pictures of the used species are now included already in the method part in the section of the fish rearing and experimental design, whereas the morphometric parameters are included in the results section.
Reviewer 3 Report
Comments to the Author:
The manuscript reports research into determination of physical meat quality parameters of Percidae and Salmonidae, important fish species in aquaculture. The subject matter is interesting and falls into the scope of the journal. Therefore, I recommend the manuscript in the journal FOODS making revision.
Line 61-62: could you explain why texture is rather questionable for fish?
Line 92: define the acronym RAS
Line 199: delete a comma after “physicochemical”, because there are two ones.
Line 199: how many measures of pH, electrical conductivity and impulse-impedance did you perform on each fish species and sample?
Line 201: when you describe the texture analyses, you should indicate the speed of the Warner-Bratzler blade during the cut of the sample. Likewise, you should indicate the parameter/s that you evaluate with this method.
Line 213: I deem that you should remove “and mean ± S.E.M. was averaged over” because this sentence is part from “Statistics” section.
Line 216: why do you use capital letter when you use the word method? For instance, this fact appears in the line 216 when you write “traditional Method”.
Lines 229-235: when you describe the evaluation of the fish, you should indicate if the measurements were performed on fresh fish or not. Likewise, you should indicate the illuminance and standard observe that you used in the color evaluation. Moreover, I consider that the expression “mean ± S.E.M.” should be removed because it is part from “Statistics” section.
Line 230: in the CIELAB system, the L* color coordinate express lightness. Thus, you should use the term “lightness” instead of “brightness” in your article.
Line 243: the acronym SEM means “standard error of the mean”. Thus, you should correct because you put “standard error” in the line 243. Likewise, I remind you that the acronyms should be defined the first time that they appear in the text.
Figure 1: in this figure appear pictures of fishes and a table. Thus, the table with the morphometric data collection should be separated from the Figure 1.
Line 266: what does “rsp.” mean?
Line 265-266: Rewrite the following sentence because this is not clear: “Mean values of the shear force measurements have been 24.70 ± 1.26 N rsp. 28.98 ± 1.16 N (Fig. 2A).”
Figures 2A and 2B: you should do these graphs bigger. Likewise, the Y axis should be divided in more parts to be clearer.
Line 296: you use “little differences”. This is not accepted, you should relate the differences with statistical terms, I mean, differences are significant or not.
Line 299: you use “Tab. 1.”. I consider that you use “Table 1” in the text, according to the instructions of the Journal. Thus, you should correct this mistake in your article.
Table 1: the following sentence should be appear below table 1: “L*: lightness (L* = 0 black and L* = 100 white), chroma a* (red-green component) and chroma b* (yellow-blue component). Means with different superscript letters differ significantly between the animal groups (p < 0.05).”. Likewise, in the same sentence you should remove the term “chroma”.
Table 1: if you use lines in order to separate the columns and/or rows, the table would be clearer.
In the discussion, firstly you discuss the texture results and after the pH. I suggest that in the “Material and methods” section you order the methods according to the order used in the “Discussion” section.
“References” section should be revised. When you use the abbreviation of the Journals, in some cases, you should use the full stop (.). For instance, “J. Food Sci.” instead of “J Food Sci”.
Author Response
Line 61-62: could you explain why texture is rather questionable for fish?
Thank you for your comment. The problem with previous measurements of the shear force is explained in the following sentences. However, it was really not easy to comprehend, so we have restructured the sentences as followed:
….The texture is one of the main quality parameters for fresh fish and is defined by its dryness, chewiness and juiciness [5]. But, the measuring methods used to determine texture, is for the most part not objective and rather questionable for fish. In the fish industry, the texture is commonly tested by the “finger method”, which to a large extent depends on the subjective evaluation of the person who does the test [6]. For that reason, Sigurgisladottir and colleagues studied already different methods for examining the texture and could show that the shear force method, used standardized for analyzing meat of mammals and birds, was found to be more sensitive than the puncture methods and best suited for practical application [7]. Nevertheless, this method is still not standardized for fish. One of the causes is the high diversity of the fish and the associated texture values…..
Line 92: define the acronym RAS
Thank you for pointing out that we have not explained some abbreviations, like RAS (Recirculating aquaculture systems). We have included this in the text.
Line 199: delete a comma after “physicochemical”, because there are two ones.
Thank you, we deleted the comma.
Line 199: how many measures of pH, electrical conductivity and impulse-impedance did you perform on each fish species and sample?
Thank you for the hint of being not detailed enough. We did include the following sentence: These three parameters were measured once for each filet of every animal at room temperature 5 min and 1 h post mortem.
ILine 201: when you describe the texture analyses, you should indicate the speed of the Warner-Bratzler blade during the cut of the sample. Likewise, you should indicate the parameter/s that you evaluate with this method.
We did include it – please see line 222.
Line 213: I deem that you should remove “and mean ± S.E.M. was averaged over” because this sentence is part from “Statistics” section.
We deleted S.E.M. in the method and described it only in the Statistic section.
Line 216: why do you use capital letter when you use the word method? For instance, this fact appears in the line 216 when you write “traditional Method”.
Thank you for this advice. We did change this.
Lines 229-235: when you describe the evaluation of the fish, you should indicate if the measurements were performed on fresh fish or not. Likewise, you should indicate the illuminance and standard observe that you used in the color evaluation. Moreover, I consider that the expression “mean ± S.E.M.” should be removed because it is part from “Statistics” section.
We have added the word “fresh” in the methods - line 229. Additionally we added: Standard Illuminant D65: Average daylight (including ultraviolet wavelength region) with a correlated color temperature of 6504K. Furthermore, we have included the missing parameters and removed the term “mean ± S.E.M.”
Line 230: in the CIELAB system, the L* color coordinate express lightness. Thus, you should use the term “lightness” instead of “brightness” in your article.
Thank you for the advice. We did change this section and explained it in more detail.
Line 243: the acronym SEM means “standard error of the mean”. Thus, you should correct because you put “standard error” in the line 243. Likewise, I remind you that the acronyms should be defined the first time that they appear in the text.
Thank you for the hint and excuse the confusion. The acronym SEM is now defined as the right term “standard error of the mean” in line 286.
Figure 1: in this figure appear pictures of fishes and a table. Thus, the table with the morphometric data collection should be separated from the Figure 1.
The pictures of the used species is now included already in the method part in the section of the fish rearing and experimental design, whereas the morphometric parameters are included in the results section.
Line 266: what does “rsp.” mean?
Please see the following comment.
Line 265-266: Rewrite the following sentence because this is not clear: “Mean values of the shear force measurements have been 24.70 ± 1.26 N rsp. 28.98 ± 1.16 N (Fig. 2A).”
Thank you for the advice. The sentence has been revised. “The shear force measurements of maraena whitefish was 24.70 ± 1.26 N and of rainbow trout 28.98 ± 1.16 N (Figure 2A).”
Figures 2A and 2B: you should do these graphs bigger. Likewise, the Y axis should be divided in more parts to be clearer.
We separated the Y-axis in more parts and increased the size of the Figure 2A and 2B.
Line 296: you use “little differences”. This is not accepted, you should relate the differences with statistical terms, I mean, differences are significant or not.
Thank you for the hint. We did revise the sentence.
Line 299: you use “Tab. 1.”. I consider that you use “Table 1” in the text, according to the instructions of the Journal. Thus, you should correct this mistake in your article.
Table 1: the following sentence should be appear below table 1: “L*: lightness (L* = 0 black and L* = 100 white), chroma a* (red-green component) and chroma b* (yellow-blue component). Means with different superscript letters differ significantly between the animal groups (p < 0.05).”. Likewise, in the same sentence you should remove the term “chroma”.
Thank you for the advice. The mistake in the usage of Journal instructions is corrected. We changed the table like suggested.
Table 1: if you use lines in order to separate the columns and/or rows, the table would be clearer.
Thank you for the advice, we did use lines now for a better visibility of the data.
In the discussion, firstly you discuss the texture results and after the pH. I suggest that in the “Material and methods” section you order the methods according to the order used in the “Discussion” section.
We have reorganized the text of Materials and Methods to make it consistent with the illustrations and the discussion.
“References” section should be revised. When you use the abbreviation of the Journals, in some cases, you should use the full stop (.). For instance, “J. Food Sci.” instead of “J Food Sci”.
We fundamentally revised the list of references and added additional references.
Round 2
Reviewer 1 Report
The manuscript has improved, but there are still issues that should be discussed and improved. Some new references have been added on studies done on fish, but the authors still think that not mushc have been done on the physical quality of fish flesh, this has been studied since the 1950's at least, but there are many newer studies that the authors have overlooked. For instance - the authors very rightly cite the review paper by Poli and coworkers from 2005 but studies on stress and rigor have also been published earlier on salmon (1997).
the statement on line 63 should be removed - this is an insult to all the good studies that have been done on fish up through the years and that the authors have chosen to ignore or not found. This also part goes for the statement on l 53/54 . this is the problem with determination of physical parameters and biological materials.
l 110 the age of the fish is not relevant for sale, but could be for quality- as size is highly related to water/fat content (at least for fatty fish).
l 216 measuring shear force 30 minpost mortem? you will influence the rigor process of the flesh by cutting in the flesh - so this is a bit questionable
water holding capacity you ues 0.3 g??? that is a very small piece and how do you ensure that this is a representative sample and 35 kg weight!! we use 2g (mince) and a low speed centrifugation - and have tested the conditions carefully
both texture and pH cannot be the most important parameter
Author Response
The manuscript has improved, but there are still issues that should be discussed and improved. Some new references have been added on studies done on fish, but the authors still think that not mushc have been done on the physical quality of fish flesh, this has been studied since the 1950's at least, but there are many newer studies that the authors have overlooked. For instance - the authors very rightly cite the review paper by Poli and coworkers from 2005 but studies on stress and rigor have also been published earlier on salmon (1997).
-->Thank you, you are right. So far, we haven’t integrated enough references regarding stress and rigor mortis into the manuscript. We hope that we have made up for it now, to your delight. Please see line 82-88.
the statement on line 63 should be removed - this is an insult to all the good studies that have been done on fish up through the years and that the authors have chosen to ignore or not found.
-->We did delete this sentence.
This also part goes for the statement on l 53/54 . this is the problem with determination of physical parameters and biological materials.
We have changed the sentence to “However, the basic methods used to check meat quality in mammals and birds are still not standard in fish due to its high species diversity”.
l 110 the age of the fish is not relevant for sale, but could be for quality- as size is highly related to water/fat content (at least for fatty fish).
--> Thank you for your comment. And yes, you are right. Age is a relevant for quality. For that reason we did mention this in the text-line 117pp: “… It is known that fish at a fully developed stage convert food not only into protein deposition but also into fat [32]. For economic reasons, fishes in aquaculture and therefore also our used fishes were used when reaching the commercial weight.”
--> In future studies we plan to investigate the meat quality in relation to size and age. In this study, however, the size and weight of the animals was decisive, as at this point in time these are the key factors for fish sale in our region.
l 216 measuring shear force 30 minpost mortem? you will influence the rigor process of the flesh by cutting in the flesh - so this is a bit questionable
--> Thank you for your comment. So we just made our measures in the pre rigor state. Comparability is given to similar handling. Next time, we will also measure post rigor state which we did mentioned already in the discussion.
“The long term examination of the pH-value including the rigor mortis will be the goal of a future study. As a part of this, the start time point of the pH value increase will be determined, reflecting the production of alkaline bacterial metabolites in spoiling fish.” (line 422-424)
water holding capacity you ues 0.3 g??? that is a very small piece and how do you ensure that this is a representative sample and 35 kg weight!! we use 2g (mince) and a low speed centrifugation - and have tested the conditions carefully
--> The Dummerstorfer Hypress method was designed to measure the holding capacity of the free water of the ground muscle sample. Mincing can destroy the cell membranes and will “dilute” the free water amount with immobilized cell water. In this method, we refer directly to the initial weight of the sample, which saves the determination of the dry matter of the sample. We minimize water evaporation with the low initial weight of 0,3 gramm and will have no meat residues in the filter paper. This method had a repeatability of 0,94 (GROSSE et al. 1975) and is well known and documented in red meat. Therefore, we are hopeful to establish it also in fish.
both texture and pH cannot be the most important parameter
-->Thank you for your comment. You are right.We deleted the word “most” in line 60.
Furthermore, we did describe that texture is important (line 373) and one of the main quality parameters for fresh fish (line 61) But in our opinion, only the pH value is the most important (line 256). “…The pH-value is by far the most important meat quality parameter in all species. It gives information about the chemical development of meat and the ripening process in mammals and poultry. This is also the case for fish.”